# Pain in Pets: Beyond Physiology

**DOI:** 10.3390/ani13030355

**Published:** 2023-01-19

**Authors:** Roberta Downing, Giorgia Della Rocca

**Affiliations:** 1The Downing Center for Animal Pain Management, Windsor, CO 80550, USA; 2Research Center on Animal Pain, Department of Veterinary Medicine, University of Perugia, 06126 Perugia, Italy

**Keywords:** companion animals, pain, analgesia, acute pain, chronic pain, maladaptive pain, clinical bioethics

## Abstract

**Simple Summary:**

Chronic pain in pets concerns pet owners and veterinarians alike. Acute pain that is not appropriately addressed can evolve into chronic (long-lasting) maladaptive pain. Despite advances in veterinary medicine, there remains a gap between pain management knowledge and its execution. Veterinary clinicians can and should embrace the foundational principles of clinical bioethics, translated from human medicine, for the benefit of their patients. Pet pain is not simply a physiologic issue. By reframing companion animal pain as a bioethical issue, as described in this paper, veterinarians affirm their commitment to closing the gap between what is known and what is done for their painful patients.

**Abstract:**

Animals do not speak a language humans understand, making it easy to believe that they do not experience pain the way humans do. Despite data affirming that companion animals can and do experience pain much as do humans, there remains a gap between companion animal acute pain management knowledge and its execution. Companion animal pain is not simply a physiological issue. Veterinary clinicians can and should embrace the foundational principles of clinical bioethics—respect for autonomy, nonmaleficence, beneficence, and justice—translated from human medicine for the benefit of their patients. By reframing companion animal pain as a bioethical issue, as described in this paper, veterinarians affirm their commitment to closing the gap between what is known and what is done for painful companion animals. This takes pet pain beyond physiology.

## 1. Introduction

“Pain is a more terrible lord of mankind than even death”—Albert Schweitzer, 1922.

The history of veterinary medicine reflects a failure to address companion animals’ acute pain, as witnessed by early textbooks [1,2]. For instance, the early use of anesthesia in animals was for the purpose of restraint, not to relieve pain [2]. The first textbook dedicated to animal pain was not published until 1992 [3]. In 1997, the first non-steroidal anti-inflammatory drug for dogs was licensed in the United States [4]. Since that time, interest in companion animal pain and the accompanying literature has exploded. Investigators have worked to identify more effective ways to treat pain in companion animals. The first companion animal pain management guidelines were published in 2007 [5], updated in 2015 [6], and updated again in 2022 [7]. Yet, despite written guidelines, textbooks, and peer-reviewed articles, many dogs and cats do not receive acute pain relief [8,9].

What is currently known about recognizing, preventing, and alleviating companion animal pain is plentiful and easily accessible. It is beyond the scope of this paper to reiterate those details. Readers are guided to review the various guidelines cited among the references. Veterinary professionals are called to advocate on behalf of beings who possess no voice of their own and who often mask their acute pain. Lloyd Davis, a veterinary pharmacologist, provided insight when he stated:

“One of the physiologic curiosities of therapeutic decision-making is the with-holding of analgesic drugs because a clinician is not absolutely certain that the animal is experiencing pain. Yet, the same individual will administer antibiotics without documenting the presence of a bacterial infection. Pain and suffering constitute the only situation in which, I believe that, if in doubt, one should go ahead and treat” [10].

Pain is more than a physiological issue. It is also a bioethical issue. Reframing companion animals’ acute pain as a bioethical issue allows veterinary professionals to embrace foundational bioethical principles on behalf of their patients, empowering the profession to close the gap between what is known and what is done to manage acute pain to prevent it from becoming chronic maladaptive pain.

## 2. Clinical Bioethics and Companion Animal Acute Pain

In *Principles of Biomedical Ethics, 8th ed.*, Thomas L. Beauchamp and James F. Childress articulate four foundational principles of clinical bioethics—respect for autonomy, nonmaleficence, beneficence, and justice [11]. Translating these bioethical principles for application to clinical veterinary practice can provide guidelines within which to reframe companion animal acute pain as more than a simple sensation. This reframing acknowledges that “our love for our pets should be shaped and informed by our recognition of the ways in which their needs and their lives are their own, particular to the sorts of animals they are” [12].

A bioethical reframing of acute companion animal pain warrants a brief review of pain science.

Pain is subjective and individual, but it must be classified to be discussed [13]. Clifford Wolf, MD, updated the conception of pain as a spectrum from adaptive/“good” pain to maladaptive/“bad” pain [14]. Predictable acute pain (e.g., surgery) provides an opportunity for pain prevention and mitigation [1,7,15,16,17,18,19]. Post-operative pain, if untreated or undertreated, can evolve into chronic maladaptive pain [14,20,21,22]. When practitioners manage acute pain appropriately, it prevents permanent anatomic changes in the nervous system that shift the landscape toward chronic maladaptive pain [21,22]. It is easier to prevent pain than to “chase” it once it takes hold in the body [1,7,14,15,16,17,18,20,21,22,23].

Humans and non-human mammals share similar nervous systems, so pain physiology is similar across species lines. Animal pain reflects the animal’s awareness of the threat to itself, encompassing both sensory and emotional dimensions [24]. Responding to companion animal pain is not simply about physiology but involves empathy for their analogous pain experience [1,25,26]. Nonhuman animals can experience chronic maladaptive pain [14,21], and this knowledge must create urgency in veterinarians to give patients the benefit of the doubt by considering how intense pain might be during a procedure. Companion animal pain is assessed by proxy, analogous to the best interest standard of decision-making in clinical bioethics [11]. Treatment decisions are made on behalf of beings who are non-verbal and whose pain experiences are influenced by the environment, breed, age, and previous experiences [18,20,23].

Companion animal pain eludes precise anatomic, physiologic, and/or pharmacologic definitions, creating barriers to aggressive, pre-emptive acute pain management. In one study, Canadian veterinarians were surveyed to determine influencing factors in the delivery of post-operative analgesic drugs [8]. The main driver for pain treatment was the veterinarian’s perception of the degree of the patient’s pain [8]. Over 50% of these veterinarians did not provide analgesics [8], yet both pain science and the principle of analogy affirm that 100% of animal patients subjected to surgery experience acute pain. A systematic review of analgesic practices for dogs undergoing ovariohysterectomy revealed that a minority of those patients received acute pain care in alignment with current guidelines [9]. Assessing and responding to acute companion animal pain remains challenging, especially with a narrow focus on physiology.

## 3. Translating and Applying Bioethical Principles to Companion Animals in Pain

Expanding veterinary perspectives about companion animal pain beyond physiology and translating bioethical principles for action means embracing animals’ moral status and moral consideration of their pain. A number of philosophers affirm animals’ moral agency [1,27,28], recognizing the complexity of their consciousness and awareness, the richness of their emotional lives [1,29,30], and their ability to express preferences [1,31,32]. Their moral status makes their pain worthy of moral consideration [1,27]. Veterinarians were once taught that concerns about animal pain were rooted in sentimentality [1]. Yet, Darwin and others have argued that the differences in mental and moral capacities between humans and non-human animals are differences of degree and not differences of kind [33], providing the necessary undergirding for moral consideration of companion animal pain [27,34,35,36,37]. Companion animal ethics has been called by some a “neglected field of bioethics” [38,39].

Veterinarians enter practice to help animals and to support the human–animal bond their clients share with their pets. They do their best to respond to patients in pain with sympathy, compassion, and empathy, which grounds a bioethical response [25,40], recognizing an animal’s preference to avoid pain. Animals live “in the moment”, potentially making pain worse for them than for humans [1,41]. They cannot deal intellectually with pain, think of better moments in the past, or anticipate that their pain will end in the future [1,41]. Veterinarians may intuitively appreciate the moral consideration of companion animals in pain, but the time has come to embrace a formal bioethical dimension to treating these patients. Translating the foundational principles of clinical bioethics—respect for autonomy, nonmaleficence, beneficence, and justice—as described by Beauchamp and Childress [11] can benefit companion animals as these principles have benefitted human patients since 1979.

Autonomy implies self-rule [11]. In human health care, autonomous decisions are made by the patient or surrogate decision-maker once that individual understands the provided information about the action in question. The intent to proceed should be free from coercion [11]. In veterinary medicine, the pet owner is the surrogate decision-maker for the companion animal patient, trying to determine what is in the pet’s best interest [11] based on the veterinarian’s recommendations. A pet’s pain adds urgency to the veterinarian’s obligation to facilitate the pet owner’s understanding of treatment options.

A companion animal in pain also needs consideration for its autonomy. Human medicine respects children’s autonomy by recognizing their expressed interests and preferences based upon their chronological and intellectual ages [42]. Parents and guardians are the ultimate decision-makers, but child patients often assent to participating actively in their own care [42,43]. Much like pre-lingual children, companion animals can and do express preferences in many ways [32,44,45], not unlike what Navin and Wasserman describe as pediatric assent [42]. Extrapolating from this concept, one aspect of treating companion animal pain is to respect the patient’s expressed preferences. Veterinarians can presume their patients’ preferences include avoiding pain. Companion animal preferences are relevant during acute pain care because the inability to express preferences about handling, etc., can lead to anxiety and fear [44,45,46]. Anxiety and fear amplify pain, and pain amplifies anxiety and fear [47].

Nonmaleficience means avoiding harm to others [11], distinct from actively pursuing good on another’s behalf (beneficence) [11]. For acutely painful companion animals, nonmaleficence demands that veterinarians manage pain effectively by adhering to current standards of acute pain management, and by reassessing painful patients frequently to continue relieving pain and preventing suffering. With respect to pet owners, applying nonmaleficence means assisting clients to discern potentially harmful misinformation that they may encounter. The Internet is filled with inaccuracies. Nonmaleficience includes protecting both pet owners and companion animal patients from negligence born of misinformation.

In contrast to avoiding harm (a passive process), the principle of beneficence demands positive action on behalf of another [11]. For companion animals in pain, beneficence guides the veterinarian to consider ways in which pain may be anticipated, prevented, and (when it cannot be prevented) treated. Beneficence demands making an appropriate pain management plan and includes understanding the pet’s expressed preferences for care and handling. It is in the painful companion animal’s best interest for the veterinarian to apply the most current standards of pain care. Beneficence toward the pet owner demands that the veterinarian communicate effectively about the pet’s condition throughout treatment. As conditions change, beneficence supports balancing ongoing acute pain care with compassionate communication and dialog with the pet owner to support the client’s ability to continue making decisions based on the pet’s best interest.

In human medicine, the bioethical principle of justice focuses on the equitable allocation of healthcare resources, setting priorities, and rationing (distributive justice), as well as access to medical research and protection of research study participants [11]. Pet ownership is voluntary, and pet care is a pain for out-of-pocket, so the allocation of veterinary medical resources is determined by the pet owner’s choices coupled with their ability to pay for recommended care.

One conception of justice is fairness, and this is the most useful and relevant translation of this principle for application to veterinary practice. Fairness for painful companion animals demands creating and providing an individualized pain management plan for each patient that reflects current guidelines, avoiding any biases that might interfere with best practices. Fairness for pet owners means providing each client with the best effort on behalf of their pet. When the pet faces acute pain, fairness demands that veterinarians engage in shared decision-making with clients, providing transparency about proposed treatments, and articulating risks and benefits, the prognosis, and expected costs. Fairness also means setting priorities among treatment options to maximize pain relief.

## 4. A Call to Action

Veterinarians are compassionate and empathic individuals who pursue the profession to help animals. Providing appropriate acute pain management to companion animals to avoid chronic maladaptive pain is in alignment with compassionate care, as is the knowledge that standards of pain care continue to evolve and change. Veterinary surgeries and diagnostic procedures are more invasive and complex than ever. The potential for unintentional under-management of acute pain is great, and under-treated acute pain can lead to chronic maladaptive pain. Because companion animals warrant moral consideration due to their pain [1,27,28], they need comprehensive attention when they experience pain. That comprehensive attention must look beyond simple physiology, reframing their pain bioethically as well. Bioethical consideration of acute companion animal pain expands the pain management landscape beyond simple physiology. This provides an important and relevant tool for closing the gap between pain management knowledge and execution, helping veterinarians prevent acute pain from evolving into chronic maladaptive pain.

## 5. Conclusions

Any opportunity to inflict acute pain increases the veterinarian’s moral obligation to manage that pain as effectively as possible to prevent the patient from developing chronic, maladaptive pain. Reframing acute companion animal pain as a bioethical issue beyond mere physiology expands the landscape of pain management and enriches the care veterinarians provide to their painful companion animal patients. Translating the foundational principles of clinical bioethics for application to companion animals provides veterinarians with a formal framework within which to consider their painful patients. Moral consideration of companion animals and their pain complements knowledge of pain physiology and adds urgency to recognizing and addressing unmanaged or under-managed acute pain. Bioethical reframing of pet pain empowers veterinarians to close the gap between what is known and what is done for painful companion animals.

“…pain is pain, whoever experiences it, and that alone is sufficient for moral obligation…”—Anglican clergyman, Humphrey Primatt [48].

## Data Availability

The data presented in this paper are readily available in the references cited.

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
