# Peer review of "Pain in Pets: Beyond Physiology"

_animals, 2023, doi:10.3390/ani13030355_

Round 1

Reviewer 1 Report

Veterinarians must provide pain and suffering relief to their patients and recognize this state in order to make an accurate diagnosis and prescribe appropriate treatment with analgesics. There is a complex method for assessing pain, as well as a wide range of aspects to animal behavior in pain.  I find this work useful, but for the sake of the readership, it would be appropriate to mention pain tolerance and pain threshold, or to make a note about pain sensitivity in companion animal species from most sensitive to least sensitive. 

There is a mistake in the line 34, please correct year "20015"

Author Response

Reviewer # 1:

Veterinarians must provide pain and suffering relief to their patients and recognize this state in order to make an accurate diagnosis and prescribe appropriate treatment with analgesics. There is a complex method for assessing pain, as well as a wide range of aspects to animal behavior in pain.  I find this work useful, but for the sake of the readership, it would be appropriate to mention pain tolerance and pain threshold, or to make a note about pain sensitivity in companion animal species from most sensitive to least sensitive. 

There is a mistake in the line 34, please correct year "20015"

Thank you for your detailed reading of this manuscript.  You have made this comment:

for the sake of the readership, it would be appropriate to mention pain tolerance and pain threshold, or to make a note about pain sensitivity in companion animal species from most sensitive to least sensitive. 

Respectfully, I rely upon the reader’s initiative to access the key pain management guidelines that are referenced.  I have clarified that this is a task that must be undertaken by the reader.  It is beyond the scope of this particular article to outline in detail these issues that you have articulated.

Thank you for noticing the extra “zero” in line 34.  Despite my very best efforts, I am afraid that a simple mistake like this eluded my review.

Reviewer 2 Report

Pain management for companion animals is a current field of research due to the ethical, physiological, and social relevance associated to animal welfare. This communication reflect the advance on this subject from a professional and owner-related point of view. I left some comments hoping they can improve the manuscript.

Lines 7-9: One of the main issues in companion animals is the specie-specific differences regarding pain recognition and management (e.g., not every drug used in dogs can be used in cats, and cats show less pain-related behavior). This might be mentioned as well.

Line 11: Please, delete the space before “Acute and chronic pet pain”

Line 14: Consider rewrite this sentence. For example, it could say that pain is a detrimental sign in companion animals that has being gaining relevance in the last years due to the close bond to humans, forcing veterinarians to improve techniques for pain assessment.

Line 17: Consider adding that pain neurobiology has similarities in most mammals.

Line 20: Please, delete the space before “Reframing companion animal”.

Simple summary and abstract: Please, state the aim of the present article.

Line 24: Consider adding “companion animal” as a keyword.

Line 27: This statement needs to include a citation of the current definition of pain according to the IASP (doi: 10.1097/j.pain.0000000000001939). And, please, add a final point at the end of the sentence.

Lines 28-29: I recommend including and state some of the factors that are associated to the failure in addressing companion animal acute pain (e.g., different species, behavioral responses, non-specific physiological parameters, etc.).  

Line 33: Please, add that assessment tools has increased the study of pain recognition (Doi: 10.1080/23144599.2019.1680044)”.

 Lines 30, 31, 35, 38, 43, 44, 47, 57: Please check the double spaces and remove them and add the missing ones. Please review these same aspects throughout the entire document.

Lines 33-35: It would be interesting to briefly include the main ideas inside the first animal pain management guidelines and what were the updates.

Line 36: It would be appropriate to mention that the lack of knowledge about these tools, the fear for drug toxicity, and the experience of the veterinarian/owner to assess and manage pain influence this effect (doi: 10.1111/j.1467-2987.2004.00175.x)”.

Line 53: Please, revise citation style.

Line 59: I recommend adding some physiological aspects of pain. For example, that pain perception can affect metabolic, neurophysiological, and behavioral responses of animals. Therefore, it is a bioethical requirement to mitigate pain.

Line 51: It would be helpful to include some lines regarding the objective of the present article so the reader can know what to expect while reading this study.

Lines 67-69: Could the authors include an example on dogs/cats where non-treated acute pain transforms into chronic pain?

Line 71-72: Which neural structures are similar between humans and non-human mammals? Please, briefly mention this since the presence of peripheral receptors for pain (or nociceptors) is important as a basis to recognize pain in companion animals and other species.

 Lines 85-86: How was evaluated the veterinarian’s perception of the degree of the patient’s pain. Please mention it.

Line 107: Change “chare” to “share”. Be mindful for other typographical errors in the manuscript, for example lines 150, 152, 154

Lines 118-125: I consider appropriate to include that the perception that owners might have regarding animal emotions can be controversial. On one hand, it can lead to anthropomorphic practices that are not beneficial for pets. However, a positive end is when owners accept that, as humans, animals can feel pain and then are more interested in welfare and medical treatments to avoid pain https://doi.org/10.3390/ani11113263

References:  Please review the references to check if they match the way the journal asks for.

Author Response

Please see my attachment...  I included more comments/responses than would be appropriate to copy into this text box.

Thank you.
